# Sex-Specific Associations of Red Meat and Processed Meat Consumption with Serum Metabolites in the UK Biobank

**DOI:** 10.3390/nu14245306

**Published:** 2022-12-14

**Authors:** Bohan Fan, Jie V. Zhao

**Affiliations:** School of Public Health, Li Ka Shing Faculty of Medicine, The University of Hong Kong, Hong Kong, China

**Keywords:** red meat, processed meat, UK Biobank, serum metabolites

## Abstract

Red meat consumption has been found to closely related to cardiometabolic health, with sex disparity. However, the specific metabolic factors corresponding to red meat consumption in men and women have not been examined previously. We analyzed the sex-specific associations of meat consumption, with 167 metabolites using multivariable regression, controlling for age, ethnicity, Townsend deprivation index, education, physical activity, smoking, and drinking status among ~79,644 UK Biobank participants. We also compared the sex differences using an established formula. After accounting for multiple testing with false discovery rate < 5% and controlling for confounders, the positive associations of unprocessed red meat consumption with branched-chain amino acids and several lipoproteins, and the inverse association with glycine were stronger in women, while the positive associations with apolipoprotein A1, creatinine, and monounsaturated fatty acids were more obvious in men. For processed meat, the positive associations with branched-chain amino acids, several lipoproteins, tyrosine, lactate, glycoprotein acetyls and inverse associations with glutamine, and glycine were stronger in women than in men. The study suggests that meat consumption has sex-specific associations with several metabolites. This has important implication to provide dietary suggestions for individuals with or at high risk of cardiometabolic disease, with consideration of sex difference.

## 1. Introduction

Cardiovascular disease (CVD) is the leading cause of mortality [1]. Diabetes is an established risk factor for CVD. Dietary factors are key components for cardiometabolic disease prevention. In recent years many national dietary guidelines, such as the US the 2020–2025 Dietary Guidelines, have recommended limiting red and processed meat due to their associated detrimental health outcomes [2]. Several observational studies have shown that red meat consumption is linked with increased risks of cardiometabolic disease, including CVD and diabetes [3]. The changes in metabolites, such as lipids and glucose, which occur prior to cardiometabolic events, are predictive of these events. Disentangling the specific metabolic factors that correspond to red meat consumption could provide insights for cardiometabolic disease prevention strategies. In addition, red meat consumption is associated with cardiometabolic deaths with a potential sex disparity; the association is higher in men than in women [4]. However, the sex-specific association of red meat consumption with metabolic factors have not been clarified. The dietary guidelines, including the US 2020–2025 Dietary Guidelines, have not considered men and women separately in the recommendations of meat intake, possibly due to the lack of relevant evidence. Prior studies suggested red meat consumption may be related to higher gut microbiota-generated metabolite trimethylamine *N*-oxide [5], lipoprotein subclasses and lipids [6], ferritin, and lower glycine [7]. However, no studies took account of a wide range of metabolites corresponding to different biological pathways and none of them considered sex disparity.

Given the apparent sex differences in meat consumption and in cardiometabolic disease risk, it is worthwhile to assess the association of meat consumption and metabolic factors in a sex-specific manner. Metabolomics, which is systematic profiling of the small circulating molecules of our body, can reveal metabolic alterations due to changes in diet and lifestyle and provide a way to comprehensively assess the associations [8]. Therefore, we systematically assessed metabolic factors corresponding to red meat and processed meat consumption in men and women with metabolomics. The identification of the sex-specific associations of red meat consumption with metabolic factors may help understand the sex-specific associations of red meat consumption with cardiometabolic diseases. The information can also be applied to health promotion program and provide more references for dietary guidelines.

## 2. Materials and Methods

### 2.1. Study Population

This study examined the sex-specific associations of unprocessed red meat and processed meat consumption with serum metabolites in the UK Biobank. The UK Biobank is a population-based cohort study consisting of half a million individuals aged 40–69 years across the UK between 2006–2010 [9]. The UK Biobank participants were registered via National Health Service for on-going follow up.

### 2.2. Metabolomic Profiling

Metabolomic profiling was conducted using high-throughput proton nuclear magnetic resonance spectrometry (NMR) on plasma samples from approximately 120,000 randomly selected participants between 2019–2020. The samples are stored at a temperature of −80 °C and go through minimal sample preparation by only adding a phosphate buffer to each sample. A sample volume of 100 μL or 350 μL is used for the analysis and the quality-control procedures detect irregularities from potential sample degradation. The measurements of all 249 metabolites, including 14 lipoproteins subclasses, fatty acids, and various low-molecular weight metabolites, such as amino acids, ketone bodies, and glycolysis metabolites and fluid balance related metabolites (albumin and creatinine), were yielded automatically at once [10]. The NMR metabolomics platform has now been applied to large-scale epidemiological studies and in clinical settings. We obtained metabolites quantified in absolute mmol/l units and standardized them before analysis.

### 2.3. Assessment of Dietary Intake

At recruitment, participants electronically signed consent forms and were invited to assessment centres for verbal interviews, physical measures, biosample collection, and completed various questionnaires. The baseline questionnaires collected information on sociodemographic (i.e., ethnicity, employment status, marital status, education, income etc.), lifestyle (i.e., diet, food consumption, physical activity, smoking and alcohol drinking etc.), as well as a dietary questionnaire [11]. The intakes of processed meat (i.e., bacon, ham, sausages, meat pies, kebabs, burgers, chicken nuggets), and unprocessed red meat (i.e., unprocessed beef, lamb/mutton, and pork) was obtained from the baseline questionnaires. The intakes of unprocessed red meat and processed meat were transformed into weekly frequency of consumption: never eaten = 0, eaten < 1 time/week = 0.5, 1 time/week = 1, 2–4 times/week = 3, 5–6 times/week = 5.5, and ≥1 time daily = 7; ‘Do not know’ or ‘prefer not to answer’ were coded as missingness.

### 2.4. Assessment of Potential Confounders

Information on age, ethnic background, smoking status, Townsend deprivation Index, education, physical activity level, smoking status and alcohol drinking status was collected by baseline questionnaires. Ethnicity was regrouped into 5 categories as: White; Asian or Asian British; Black or Black British; Chinese; mixed; and Other. Acquired qualifications including ‘College or University degree’, ‘A levels/AS levels or equivalent’, ‘O levels/GCSEs or equivalent’, ‘CSEs or equivalent’, ‘NVQ or HND or HNC or equivalent’, ‘Other professional qualifications’ were regrouped as ‘with college/university degree’ and ‘without college/university degree’. Townsend deprivation Index is calculated based on unemployment, overcrowded households, households without car ownership, and non-home owners percentages [12]. Physical activity measurement was taken from the International Physical activity questionnaire at baseline and was classified as low, moderate, and high levels [13]. Smoking status and alcohol drinking status were classified as never, previous, and current users. Body mass index (BMI) is calculated by dividing weight (kg) by standing height^2 (m^2^) measured during the baseline assessments according to standard protocol.

### 2.5. Statistical Analysis

We used multivariable regression to assess the association of unprocessed red meat consumption and processed meat consumption with each metabolite in men and women separately, controlling for age, ethnicity, Townsend deprivation Index, education, physical activity level, smoking status, and drinking status in model 1. As BMI is a potential confounder or a mediator, we further controlled for BMI in model 2. To account for multiple testing, we controlled the false discovery rate (FDR) at 5% [14]. We then selected metabolites significantly associated with red meat consumption in men and women separately, after controlling for multiple testing. For the metabolites which passed the statistical significance in both men and women, we also tested whether the associations differ by sex. Sex differences were estimated using z-test with a statistical significance of *p* < 0.05. All statistical analyses were conducted using R version 4.0.1 (Foundation for Statistical Computing, Vienna, Austria).

## 3. Results

Baseline characteristics of the included ~79,644 participants (47.7% men) with information of both meat consumption and metabolites measures were shown in Appendix A. The mean ages of men and women were 56.7 years, and 56.3 years, respectively. As expected, men consume more meat (2.29 times per week) than women (1.98 times per week), and both men and women eat more unprocessed red meat than processed meat (Appendix A).

Among 167 metabolites examined, we found 148 significant metabolites in men and 149 in women associated with unprocessed red meat consumption (Figure 1). Majorities of associations remained after controlling for BMI in model 2 (Figure 1 and Figure 2). Among these 148 metabolites in men and 149 metabolites in women, 135 metabolites are shared in men and women. Regarding processed meat consumption, 148 significant metabolites were found in men and 130 in women (Figure 2). Among these, 122 metabolites are shared in men and in women. When testing the heterogeneity by sex, 34 metabolites had sex differences (testing for sex difference with significance of *p* value < 0.05) corresponding to unprocessed red meat consumption (Appendix A), and 45 for processed meat consumption (Appendix A).

For unprocessed red meat consumption, its positive association with branched-chain amino acids and various lipid measures in lipoprotein subclasses including cholesterol and cholesterol esters in very small very-low-density lipoprotein (VLDL), triglycerides, cholesterol, cholesterol esters, and phospholipids in medium low-density lipoprotein (LDL) and free cholesterol in medium VLDL; as well as its inverse association with glycine, high-density lipoprotein (HDL) size, cholesterol, cholesterol esters, phospholipids, free cholesterol and lipids in large HDL and all lipid measures in large HDL is larger in women than in men (Figure 1). More unprocessed red meat consumption is also related to higher apolipoprotein A1, creatinine, monounsaturated fatty acids, triglycerides, and lipids in chylomicrons and extremely large VLDL, as well as triglycerides in small LDL, medium LDL, intermediate-density lipoprotein (IDL), LDL, large LDL, and very large HDL. These metabolites have stronger associations with unprocessed red meat consumption in men than in women (Figure 1).

More processed meat consumption is associated with higher branched-chain amino acids, tyrosine, glycolysis related metabolite lactate, inflammation marker glycoprotein acetyls, triglycerides in small HDL and small VLDL, total lipids, phospholipids, free cholesterol, cholesteryl esters, cholesterol in large VLDL, and large VLDL particles sizes but lower glutamine, glycine, LDL and HDL participle sizes, phosphoglycerides, free cholesterol, cholesteryl esters and cholesterol in medium HDL, and phospholipids in HDL, large HDL, large VLDL, and very large HDL, the associations are generally larger in women than in men (Figure 2).

## 4. Discussion

This is the first study comprehensively evaluating the sex-specific association of red meat and processed meat consumption with 167 metabolites in the UK Biobank. We added to the limited evidence on sex disparity in association, by showing various metabolites, such as branched-chain amino acids, glycine, monounsaturated fatty acids, lipids in lipoprotein subclasses, apolipoprotein A1, and creatinine were linked to red meat consumption. These sex differences may underlie differential metabolic and cardiovascular risk.

Red meat is rich in essential amino acids, consistently we found red meat and processed meat consumption are associated with higher branched-chain amino acids in both sexes. Previous evidence has shown that branched-chain amino acids may link to higher risk of diabetes [7] and cardiovascular disease [15]. Their stronger positive associations in women are consistent with a previous cohort study suggesting women had a higher risk of ischemic heart disease associated with red meat consumption than men [16]. We also found the inverse associations with glycine in response to both red meat and processed meat consumption are larger in women than in men. Glycine is related to insulin resistance and may be linked to risk of type 2 diabetes [7]. The evidence regarding sex-specific associations with tyrosine and glutamine has not been reported previously, hence, these results need to be replicated.

Cholesterol and triglycerides are the major lipids that are transported in plasma by lipoproteins. The associations of red meat consumption with higher lipids for lipoprotein subclasses are partly consistent with a previous Chinese study [6], that reported increased cardiovascular disease risk with these metabolites. Our findings showed these associations of lipids and lipoprotein metabolites in response to meat consumption were stronger in women than in men. The sex disparity in apolipoprotein A1, the main apolipoprotein of plasma HDL, was also consistent with the patterns for HDL and its subclasses. However, the underlying mechanisms are unclear. A possible explanation is that red meat may alter sex hormones which could play a role modulating lipid and lipoprotein metabolism [17].

Moreover, we found red meat was associated with higher serum creatinine in men than in women. Men have more red meat consumption and more muscle mass that produce creatine than women, which were converted to creatinine that is excreted by the kidneys into the urine [18]. This is consistent with the sex-specific association of red meat consumption with creatinine. Women had more pronounced increase in glycolysis-related lactate with regard to processed meat but not red meat intake, that could be due to the use of lactate in processed meat for antimicrobial and flavoring purposes [19]. Glycoprotein acetyls biomarker could predict risk of cardiovascular and diabetic risk outcomes [20]. Here we showed more elevated inflammation marker glycoprotein acetyls associated with processed meat consumption in women than in men, possibly because processed meat could change female sex hormone estrogen that regulates metabolic inflammation [21]. Whether this biomarker mediates the sex-specific association between processed meat consumption and cardiometabolic risks requires further investigation.

Our study takes advantage of powerful metabolomics data in UK Biobank to study sex-specific associations of meat consumption and metabolic factors. Despite the novelty, there are several limitations. First, self-reported questionnaires for ascertaining dietary intake are susceptible to recall bias. Second, residual confounding may still exist, even if potential confounding factors such as baseline sociodemographic factors, lifestyle factors, and BMI were adjusted in multivariable models. These associations should be confirmed in interventional studies. Third, UK Biobank participants are not fully representative because they are healthier than the general population [22], and results from Europeans may not be generalized to other populations with different dietary habits.

Taken together, our study is in line with the US 2020–2025 Dietary Guidelines regarding the limitation of meat consumption. In addition, our study provides information on the sex-specific associations of red meat consumption with metabolites, which may promote the consideration of sex disparity in the future dietary recommendations. Although in dietary recommendations, red meat is grouped under the category of “protein foods”, in our study, a large variety of metabolites including lipids, lipoproteins, fatty acids, amino acids and fluid balance indicator (creatinine), correspond to red meat consumption. This means that people with hyperlipidemia and metabolic dysfunction, such as people with renal failure and Maple syrup urine disease (with excessive accumulation of branched-chain amino acids in the organism) are especially recommended to limit red meat intake. Our finding regarding the sex-specific associations of red meat consumption with metabolites may also be applied to health promotion programs, to advance the lowering of red meat consumption thereby lowering the risk of cardiometabolic diseases (Figure 3).

## 5. Conclusions

We showed meat consumption was associated with metabolites by sex, which are consistent with differential cardiometabolic disease risk related to meat consumption. The study has important implications in primary prevention and treatment because diet is modifiable that can be incorporated into people’s daily life. Our study suggested that people with hyperlipidemia and metabolic dysfunction, such as people with renal failure and Maple syrup urine disease, are especially recommended to limit red meat and processed meat consumption. Given the sex-disparity in the associations with metabolites, our findings also suggested that sex disparity should be considered in dietary suggestions.

## Figures and Tables

**Figure 1 nutrients-14-05306-f001:**
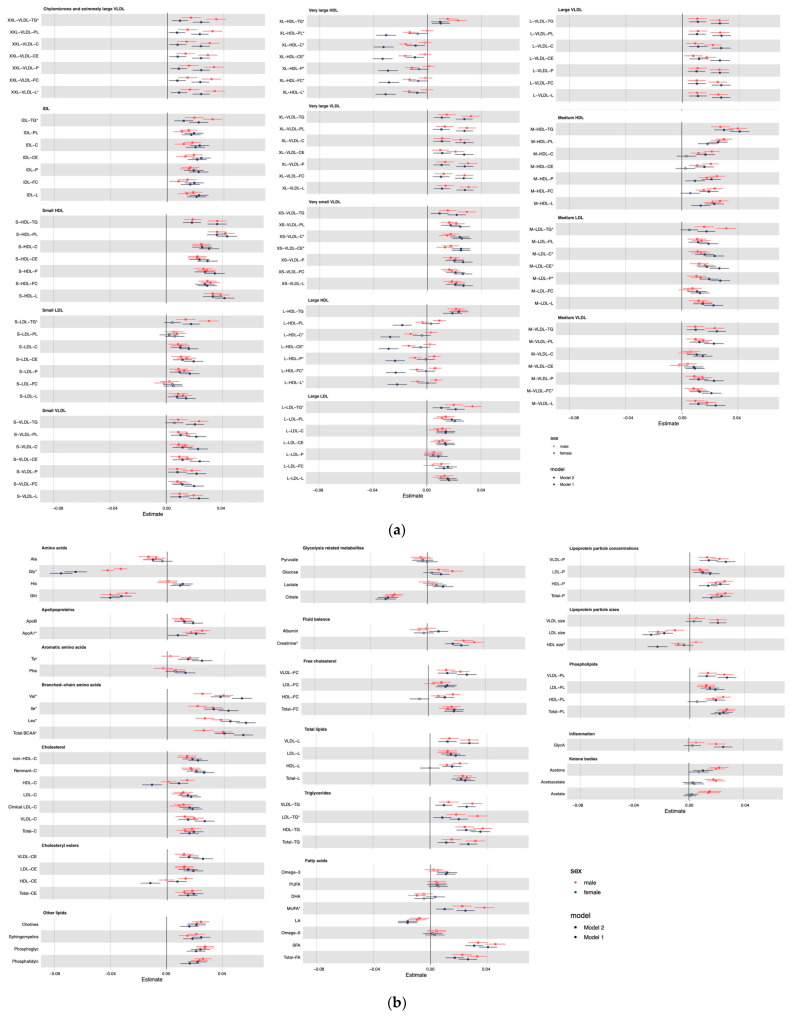
Associations of unprocessed red meat consumption with metabolites in men and women. (**a**) Associations with lipid subclasses. (**b**) Associations with other metabolites. Footnote: Mode l adjusted for age at recruitment, ethnicity (White/Asian or Asian British/Black or Black British/Chinese/Mixed/Other), Townsend deprivation Index (TDI), education (with/without university degree), physical activity level (low/moderate/high), smoking status (never/previous/current), alcohol drinking status (never/previous/current); model 2 additionally adjusted for body mass index (BMI). Non-significant associations that did not pass FDR < 5% threshold is shown in hollow dot. Metabolites showing significant sex differences after adjusting for minimally adjusted confounders are indicated with *.

**Figure 2 nutrients-14-05306-f002:**
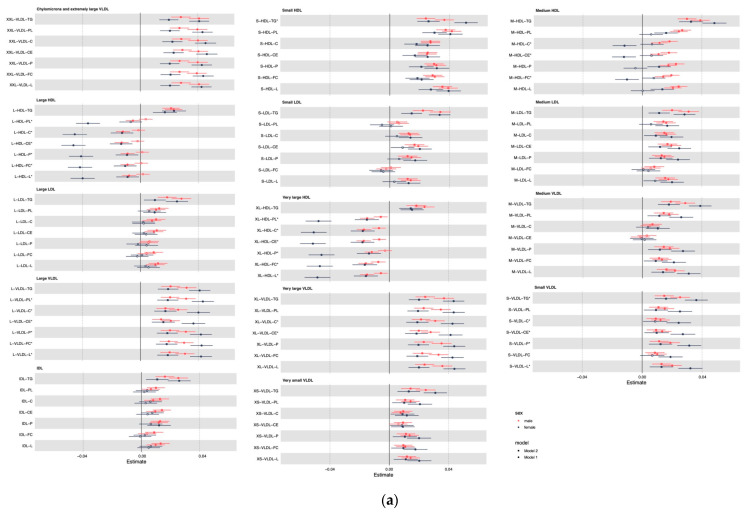
Associations of processed meat consumption with metabolites in men and women. (**a**) Associations with lipid subclasses. (**b**) Associations with other metabolites. Footnote: Mode l adjusted for age at recruitment, ethnicity (White/Asian or Asian British/Black or Black British/Chinese/Mixed/Other), Townsend deprivation Index (TDI), education (with/without university degree), physical activity level (low/moderate/high), smoking status (never/previous/current), alcohol drinking status (never/previous/current); model 2 additionally adjusted for body mass index (BMI). Non-significant associations that did not pass FDR < 5% threshold is shown in hollow dot. Metabolites showing significant sex differences after adjusting for minimally adjusted confounders are indicated with *.

**Figure 3 nutrients-14-05306-f003:**
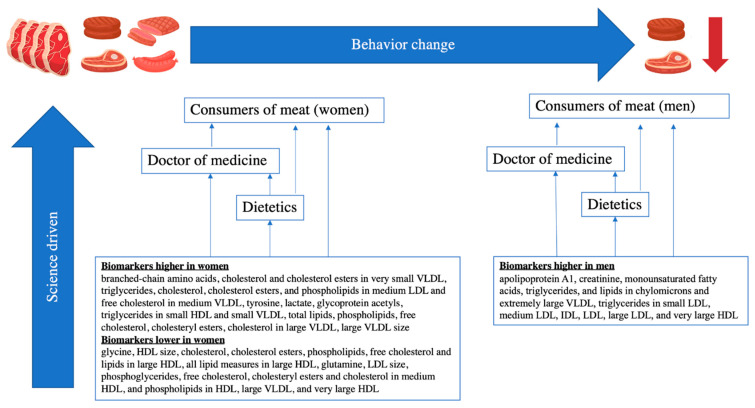
A simple model comprising the key findings of this study, consumers of meat (men and women), doctors of medicine, and dietetics to demonstrate science-based health education strategy.

## Data Availability

Data described in the manuscript will be available upon request and approval by the UK Biobank (https://www.ukbiobank.ac.uk/enable-your-research/apply-for-access, accessed on 10 November 2022).

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
