# Peer review of "Sex-Specific Associations of Red Meat and Processed Meat Consumption with Serum Metabolites in the UK Biobank"

_nutrients, 2022, doi:10.3390/nu14245306_

Round 1

Reviewer 1 Report

The article entitled “Sex-specific associations of red meat and processed meat consumption with serum metabolites in the UK Biobank” provides information about the influence of red meat and processed meat consumption on serum metabolites, taking sex into account. I believe that the article provides new information, which, moreover, is presented clearly and adequately. Additionally, the indicated bibliography is current and relevant to the topic addressed. For all these reasons, I believe that this document should be considered for publication in Nutrients, after addressing the following minor revisions.

Abstract

Page 1: In accordance with the Nutrients authors guide, remove the headlines "Background", "Methods", "Results", and "Conclusions" from the abstract.

Keywords

Page 1: Remove the green underline from the ";".

Material and Methods

Page 2: Would it be possible to briefly summarize the methodology and applications of the metabolomic platform followed and not just refer to a reference?

Page 3: Indicate the meaning of the initials "IMB " the first time you indicate it in the text.

Results

Page 4 and 5: If possible, improve the quality of the Figures as they are not legible. In the figure caption, add a period (.) after "with *". Moreover, the figure footnote with the information provided should appear in both Figure 1 and Figure 2.

Discussion

Page 7: Remove the existing enter between "and" and "BMI".

Author Response

We have uploaded the word file of point-to-point response to Reviewer 1's comments below.

Reviewer 2 Report

The problem undertaken by the authors concerns the issue of the negative aspects of red meat consumption.

It is an exciting attempt to recognize if specific metabolic factors correspond to unprocessed and processed red meat consumption in men and women. 

In conclusion, we find that meat consumption has sex-specific associations with several metabolites, and it can provide dietary suggestions for individuals with or high risk of cardiometabolic diseases. The question is who and how dietary recommendations can offer. How - by limitation of meat consumption which is in line with the US 2020-2025 Dietary Guidelines.  

Therefore the question is about the goal of the manuscript. Is it that the specific metabolic factors that correspond to red meat consumption could provide insights for cardiometabolic disease prevention strategies, or how to use this knowledge to protect individuals against cardiovascular disease by lowering red meat consumption – changes in diet and lifestyle?

The authors' conclusion that individuals with or at high risk of cardiometabolic diseases may need to limit red meat and processed meat consumption is rather general. 

Preparing a simple model like the logistic chain would be fascinating. A model comprises consumers of meat (men and women separately), doctors of medicine, and dietetics showing the most critical indicators for proper treatment to lower the risk of cardiometabolic disease.

Author Response

We have uploaded the word file of point-to-point response to Reviewer 2's comments below.

Round 2

Reviewer 2 Report

Thank you very much, the Authors, for your excellent work. I am satisfied with your improvements to the text, which now looks perfect.